# Deep Neural Network-Based Design of Planar Coils for Proximity Sensing Applications

**DOI:** 10.3390/s25144429

**Published:** 2025-07-16

**Authors:** Abderraouf Lalla, Paolo Di Barba, Sławomir Hausman, Maria Evelina Mognaschi

**Affiliations:** 1Department of Electrical, Computer and Biomedical Engineering, University of Pavia, Via Ferrata 5, 27100 Pavia, Italy; paolo.dibarba@unipv.it (P.D.B.); eve.mognaschi@unipv.it (M.E.M.); 2Institute of Electronics, Lodz University of Technology, Al. Politechniki 8, 93-590 Lodz, Poland; slawomir.hausman@p.lodz.pl

**Keywords:** magnetic field, planar coils, deep learning, CNN-based coil geometry models

## Abstract

This study develops a deep learning procedure able to identify a planar coil geometry, given the desired magnetic field map. This approach demonstrates its capability to discover suitable coil designs that produce desired field characteristics with high accuracy and efficiency. The generated coils show strong agreement with target magnetic fields, enabling manufacturers to achieve simpler structures and improved performance. This method is suitable for inductive proximity sensors, wireless power transfer systems, and electromagnetic compatibility applications, offering a powerful and flexible tool for advanced planar coil design.

## 1. Introduction

In electromagnetics, several applications depend heavily on the operation of planar coils as crucial components. The incorporation of these coils in printed circuit board (PCB) platforms provides both minimal spatial resolution and reduced physical size. The coils require optimized magnetic field distributions combined with inductive properties, which need precise modeling to achieve their target application performance.

The precise development and evaluation of planar coils strongly rely on complete knowledge of their generated magnetic fields. The calculation of magnetic fields at space points from current-carrying coils is usually performed by traditional numerical methods like, e.g., the Finite Element Method (FEM) and Boundary Element Method (BEM). The precise nature of these physics-based approaches faces multiple barriers during usage:Computational Complexity: Magnetic field computation cost increases dramatically when analyzing complex coil shapes. The modeling process faces significant complexity when dealing with detailed simulations of multilayer planar coils.Resolution vs. Efficiency Trade-Off: High spatial field analysis resolution requires substantial computing resources that usually become a barrier to real-time design implementations.Numerical Stability Challenges: The complex nature of coil patterns intensifies simulation instabilities, which eventually produces inaccurate field assessment results. The situation grows worse when working with high-frequency effects and extremely fine discretization.

The application of deep learning technology brings a transformative resolution at which to determine data processing challenges appearing in typical procedures. Complex data patterns can be identified efficiently by deep neural networks when processing relationships between data points [1]. Correct and swift planar coil geometry predictions exploit multiple magnetic field distributions through the use of reliable physics-based training data by convolutional neural networks (CNNs). The system achieves various beneficial results upon the implementation of this approach. This integration brings multiple positive effects:(a)Real-time prediction becomes possible because CNNs deliver faster computational times than those achievable when executing numerical calculations directly.(b)The trained model completes complex multiparameter optimization tasks without increasing computational needs.(c)The framework gains enhanced user understanding through added visualization components, which enable designers to make better decisions.(d)Hybrid methods demonstrate flexibility to work with multiple coil geometric shapes and design configurations because of their adaptable nature.

This paper presents a method that combines several CNNs to enhance the design of planar coils. The first goal was to construct one large training dataset. This dataset contains various designs of planar coils with their fields, calculated by means of the analytical approach and labeled with the corresponding planar coil geometry parameters. After the training phase, the CNN can accurately predict planar coil geometries and their inductance values (using follow-up calculations). Two major industrial tools are used to assess the field calculation: Simcenter MAGNET (version 2022.1) [2,3] for the field distribution and, secondly, the Texas Instruments WEBENCH Planar Coil Designer [4] for validating inductance values.

This proposed solution leverages existing tools with new capabilities that generate predictions of planar coil shapes and includes real-time design modifications. It serves as an efficient and potent design tool for winding applications, which enables its acceptance into electromagnetic systems that use advanced technologies. This framework lets users easily change coil design specifications for different manufacturing needs to achieve precise and reliable results. Its fast computation makes this tool highly effective for repetitive product development with guaranteed accuracy.

Several electromagnetic systems implement planar coils because these components conduct essential applications in proximity sensing and wireless power transfer, and their applications are expanding even to medical and biomedical fields.

Beyzavi et al. (2008) [5] focused on the analytical modeling and parametric optimization of planar microcoils for MEMS and microsystems. Their study systematically explored the influence of design parameters such as coil turns, trace width, and spacing on electrical and magnetic performance metrics, providing a structured foundation for coil design. While their approach relies heavily on deterministic modeling and optimization, our proposed deep learning framework departs from their work by learning complex relationships from field data. Kohlmeier et al. [6] gives an analysis of different technologies used to make microcoils for small actuator systems. They compare several fabrication techniques and explain how each one affects the final shape, size, and efficiency of the coils. Their work is important for understanding how the production process can change the coil’s performance. Gu et al. [7] investigate how different core materials influence the detection performance of planar coil particle sensors. They focus on optimizing design parameters and comparing detection signals from coils with various magnetic cores. Their work is valuable because it highlights the role of material selection in improving sensor sensitivity. Lope et al. [8] study printed circuit board (PCB) multi-track coils for domestic induction heating applications. They design and test coil layouts that improve heating performance while using standard PCB manufacturing. Their work shows how geometry and track design affect efficiency in heating systems. Lee et al. [9] explore the design of a printed spiral winding inductor that works well over a wide frequency range. Their work focuses on improving the inductor’s performance by carefully adjusting the winding structure, which is important for power electronics applications. Their study shows how the layout of the coil can change its electrical behavior across different frequencies. In our research, we pay attention to coil shape, but instead of using fixed formulas, we use deep learning to learn the best designs from magnetic field data. This makes it easier to find good coil shapes for different needs without many manual tests. Moreton et al. [10] utilized finite element analysis (FEA) along with experimental trials to analyze how different planar coil geometries affect sensitivity and inductance in displacement sensing, offering valuable observations but lacking predictive modeling using deep learning. Park et al. [11] developed planar coil geometries in biomedical and wearable devices, emphasizing the quality factor and coupling, though without integrating data processing frameworks (CNNs). Gołębiowski et al. [12] applied numerical simulations to MEMS planar coils for magnetic flux measurements but did not apply any predictive learning models. Our work uses the Biot–Savart law for training a deep neural network (DNN) architecture for both the accurate prediction and optimization of planar coil behavior in proximity sensing applications. This combination of physics-based modeling and data-driven learning distinguishes our approach from the existing literature and enhances its applicability in automated sensor design. Benazzouz et al. [13] developed an approach utilizing YOLOv9 and OpenCV to determine the inductance in planar coils. This approach presents an innovative technique combining computer vision and machine learning to estimate the inductance values of planar coils directly from their visual representations.

This work opens new possibilities for optimizing planar coil geometries in several technologies, such as inductive proximity sensing, wireless power transfer, and high-frequency communication systems.

## 2. Workflow for Planar Coil Geometry Prediction and Inverse Problem Solving

The illustrated workflow contains an organized approach for planar coil geometry prediction. The methodology is arranged into various stages which handle separate problems of coil geometry prediction. The workflow map is illustrated in Figure 1.

### 2.1. Data Generation

A total of 21,100 RGB images make up the workflow input, which shows magnetic field distributions produced by planar coils. The analytical model based on the Biot–Savart law produces synthesized images to determine planar coil magnetic field distributions. The geometry parameters are

▪Diameter (mm);▪Wire thickness (mm);▪Wire spacing (mm);▪Number of turns;▪Number of layers.

The analytical model is used for generating the magnetic field images through which coil parameter variability finds complete representation.

### 2.2. Deep Learning Models for Parameter Prediction

The methodology’s main component contains four convolutional neural networks (CNNs) for detecting particular magnetic field images representing coil geometry parameters. Training takes place on partial records from the generated data with the following sequence:

CNN1—Current Prediction: This network is trained with 10,000 RGB images and determines the current coil values, given the magnetic field configuration.

CNN2—Diameter Prediction: This network is trained with 10,000 RGB images to forecast coil diameter, given the magnetic field distribution.

CNN3—Turn Prediction: This network is trained with 1000 RGB images for predicting the number of coil turns, given the magnetic field distribution.

CNN4—Layer Prediction: This network is trained with 100 RGB images to determine the coil’s number of layers, given the magnetic field distribution.

## 3. Comparative Analysis of Analytical Magnetic Field Calculations and Numerical Simulations

### 3.1. Modeling of Planar Coils

The model used in this work produces a planar coil with multiple-layer design to be used along with the Biot–Savart law to generate the magnetic field distribution. The system establishes accurate coil positions through mathematical calculations relying on wire width along with spacing measurements between turns and the total turn count input. The model is as follows:(1)$$x_{i}=r_{i}\cdot cos(r_{c}\theta_{i})$$(2)$$y_{i}=r_{i}\cdot sin(r_{c}\theta_{i})$$
where $x_{i}$ and $y_{i}$ are coil points inserted into an array named coil_points:(3)$$\text{coil\_points}=\{(x_{i},y_{i})|i=0,1,2,\ldots ,res-1\}$$

The variable res represents the number of points (planar coil resolution), which defines how many coil points are used to define the spiral coil. For instance, by choosing res=360, we will obtain a single point for each 1° spiral step. The parameter $r_{c}$ could take the values 1 for clockwise and −1 for counter-clockwise.

#### 3.1.1. Input Parameters

In Figure 2 the coil parameters are shown. They are as follows:
The diameter measurement for the coil outer layers, denoted by
d_out (in meters).The thickness of each coil turn, measured through the variable w (in meters).The self-inductance equation requiring the spacing parameter s that represents the separation distance between individual turns (in meters).The total number of turns in the coil N.

#### 3.1.2. Constants and Calculations

▪Inner Diameter $(d_{in}):$ The inner diameter for a spiral coil is derived by subtracting the cumulative thickness of the windings from the outer diameter (see Figure 3) [14]:




(4)
$$d_{in}=d_{out}-2\times (w+s)\times N.$$



▪Average Diameter (d_avg): The effective diameter for inductance calculations is the arithmetic mean of the outer and inner diameters:



(5)
$$d_{avg}=\frac{d_{out}+d_{in}}{2}$$



▪Fill Ratio (ρ): The fill ratio quantifies how densely the coil is wound:



(6)
$$\rho =\frac{d_{out}-d_{in}}{d_{out}}$$



Planar coil design heavily depends on the fill ratio parameter because it determines the level of coil winding density. The quality factor (Q) becomes better when using high fill ratios because the turns show enhanced coupling that results in reduced resistance along with increased inductance [15]. Wireless power systems and high-frequency sensing benefit from low power loss because planar coils require a more than 30% fill ratio to optimize their Q factor. With the aim of performing well, the fill ratio is kept above this value.

#### 3.1.3. Magnetic Field from Each Current Element

The magnetic field contribution from each current element is calculated using the Biot–Savart law [16]:(7)$${\Delta \vec{B}}_{i}=\frac{\mu_{0}}{4\pi }I\frac{\left(\Delta {\vec{l}}_{i}\times {\vec{r}}_{i}\right)}{{\left|{\vec{r}}_{i}\right|}^{3}}$$

The total magnetic field B is the sum of the contributions from each current element, with the $B_{z}$ component being considered the most significant:(8)$$B_{z}=\sum_{i=0}^{n-2}\frac{\mu_{0}}{4\pi }I\frac{\left(\Delta {\vec{l}}_{i}\times {\vec{r}}_{i}\right)}{{\left|{\vec{r}}_{i}\right|}^{3}}$$
where ▪$B_{z}$ is the magnetic field at the observation point (T);▪$\mu_{0}$ is the permeability of free space (H/m);▪I is the current flowing through the coil (A);▪$\Delta {\vec{l}}_{i}$
is the vector representing the current element (the displacement between consecutive points
$r_{i+1}$ and $r_{i}$);▪${\vec{r}}_{i}$ is the vector from the current element
$\Delta {\vec{l}}_{i}$ to the observation point;▪$\left|{\vec{r}}_{i}\right|$ is the magnitude (distance) from the wire element to the observation point.


In modeling spiral coils, it is important to approximate the continuous winding as a series of small straight-line segments. This approach simplifies calculations of the magnetic field and related properties.

To achieve this, one point is taken for every degree of winding, resulting in 360 points per full turn. Each segment is formed by connecting two consecutive points. This method ensures an accurate representation of the coil shape, especially in areas where the magnetic field changes rapidly.

Using this level of discretization is essential for obtaining precise results, particularly when analyzing fields close to the coil or when multiple coils interact closely. Coarser discretization can lead to significant errors.

Dividing the spiral coil into straight-line segments with one point per degree provides a practical and effective way to model and analyze its electromagnetic behavior with good accuracy. This set of equations defines the 360 points along the spiral coil, with one degree per coil point.

Figure 4 illustrates the spiral coil geometry represented at different levels of discretization, demonstrating how varying the number of segments affects the approximation of the coil winding. Here, R represents the outer radius, w the wire width, s the wire spacing, and n the number of turns.

Figure 4a represents 1 degree per coil point (360 segments per turn). This highest resolution provides a very detailed and smooth representation of the spiral coil. The segments closely follow the coil’s continuous curve, capturing fine geometric details. This level of discretization is ideal for precise magnetic field calculations.

Figure 4b represents 30 degrees per coil point (12 segments per turn). With fewer points, the coil shape becomes more polygonal but is still reasonably accurate. This discretization reduces computational complexity while maintaining good approximation for many practical purposes.

Figure 4c represents 45 degrees per coil point (eight segments per turn). At this level, the coil appears as a rough polygon with noticeable straight edges between points. While faster to compute, this coarse discretization can lead to less-accurate field estimations, especially near regions with rapid changes.

Figure 4d represents 90 degrees per coil point (four segments per turn). The coil has a square shape rather than a spiral. This very rough approximation loses most of the coil’s curved shape and can cause significant inaccuracies in the analysis, mainly a low Q factor.

### 3.2. Methodology

The analytical model was implemented in a Python script, which allows for efficient and accurate calculations of the B fields. This script utilizes Equations (1) to (8) to model the system’s behavior, enabling quick analysis and predictions based on input parameters.

The analytical model in Python required validation of its magnetic field calculation accuracy through comparison with the results derived from Simcenter MAGNET (version 2022.1) simulation. The accuracy of analytical calculations was verified through tests of multiple coils.

The analytical representation of magnetic field B along the *z*-axis of the planar coil required parameters such as the coil radius, number of turns, current strength, number of layers, wire spacing, and wire diameter.

A systematic procedure was used to change planar coil parameters so that we could study their effects on magnetic field distributions, which led to the collection of a large dataset for neural network training and validation.

The procedure used identical coil setups in Simcenter MAGNET to generate results, which were then processed for direct comparison against calculated values. An evaluation of methodological concordance occurred through analysis comparing the data from Simcenter MAGNET simulations and the data produced by the Python-based analytical model.

#### 3.2.1. Data and Configurations

The following configurations were analyzed:▪**Dataset with current as a parameter:**

Fixed radius of 3 mm, 1 turn, and varying currents of 1 A, 0.8 A, 0.6 A, and 0.4 A.

▪
**Dataset with radius as a parameter:**


Currents fixed at 1 A and single-turn coils with radii ranging from 2 mm to 3 mm.

#### 3.2.2. Analytical Implementation

Using Python, the software tool produced calculations for magnetic field strength measurements throughout the *z*-axis of the planar coil from a −5 mm to 5 mm distance with the specified coil geometry. Through the Biot–Savart law, the script determined how current elements contribute to a given magnetic field value. The evaluation incorporated the following set of parameters into the mathematical calculations:

#### 3.2.3. Key Parameters

▪Current dependency analysis through values ranging from 0.4 A to 1 A.▪Triangle coils exhibit a 3 mm radius value for their effective diameters because their physical size measures 6 mm.▪The planar coil in the code in Python contains 9 turns, equal to the one used in the real planar coil design of Simcenter MAGNET (version 2022.1).▪Wire thickness: 4 mils (0.1016 mm), consistent with the physical dimensions of the simulated coil simulated in Simcenter MAGNET.▪The turns: 4 mils (0.1016 mm), the distance between turns to achieve uniform spacing of all turns.

**Config 1:** coil diameter—9 mm; wire width—4 mils (0.1016 mm); wire spacing—4 mils (0.1016 mm); number of turns—9 turns; number of layers—8.

**Config 2:** coil diameter—9 mm; wire width—3 mils (0.076 mm); wire spacing—3 mils (0.076 mm); number of turns—28 turns; number of layers—8.

The developed designs included a two-dimensional planar coil and three-dimensional geometric representation of stacked planar coils which served to model magnetic field behavior and test the computational algorithm framework. This study also developed visual representations of the two planar coil designs, which are depicted in the presented figures (see Figure 5).

The single-layer spiral coil exists as a 2D planar design, as shown in Figure 5a. The 9 turns of the coil extend to a diameter of 9 mm. A 4-mils (0.1016 mm) distance between all coil turns is achieved thanks to its defined wire width and wire spacing of 4 mils (0.1016 mm). Such a configuration generates a uniform magnetic field direction along the *z*-axis because of its symmetrical layout. When used for planar inductive sensing or PCB-integrated designs, this configuration stands out as the optimal option. A single-layer structure provides an essential validation environment to test analytical alongside simulation models.

Three-dimensional planar coils feature the two-dimensional design element with distributed layers placed on the *z*-axis, as shown in Figure 5b. The helical coil configuration reaches a total stacking height of 0.816 mm because of the design. A duplicated two-dimensional design consisting of ten concentric turns exists across all layers on the three-dimensional structure with a maintained wire width and spacing as well as circular bounding radius. Each uniform interlayer distance creates a uniform overall structure, creating a magnetic field that is superposed on the overall magnetic field strength. Stacked-layer modeling is useful in applications that need elevated magnetic field performance with enhanced coupling efficiency such as wireless power transmission and precise sensing operations.

### 3.3. Coil Design in Simcenter MAGNET

Figure 6 presents the coil model as implemented in Simcenter MAGNET, along with its physical layout. It illustrates key aspects of the model, and several important features can be noted:**Current Direction:** The current flows in alternating directions between the layers, as shown by the arrows in Figure 6. Positive current flows upward, while negative current flows downward, creating a layered distribution of current throughout the coil structure.**Turn Configuration:** The coil is designed with uniformly spaced turns, which helps ensure a symmetric magnetic field distribution along the *z*-axis. This symmetry is important for achieving stable and consistent sensing performance.**Coordinate System:** In the simulation setup, the *z*-axis is defined as the direction of interest for observing the magnetic field, while the x and y axes lie in the plane of the coil.**Simulation Environment:** The coil was simulated in Simcenter MAGNET by accurately setting its geometric and material properties, along with the input current values. These parameters reflect the real physical characteristics of the coil to ensure meaningful and realistic simulation results.

### 3.4. Results and Analysis

The computational magnetic field results matched the output of Simcenter MAGNET simulations. Analytical values matched simulated results across all configurations, thus proving the high ability of the developed model in generating real magnetic fields.

A detailed comparison between the analytical results and the simulation data obtained from Simcenter MAGNET is presented in Figure 7. The subplots illustrate the magnetic field magnitude along the *z*-axis for several configurations, where both the coil current and radius were varied. These variations were chosen to evaluate the accuracy and consistency of the analytical model under different physical conditions.

The results show that the analytical calculations and simulation outputs follow the same overall trend. Although some differences are observed in the absolute values, the general shape and behavior of the magnetic field are in strong agreement, demonstrating the reliability of the model.

Two main observations can be highlighted:**Effect of Current Variation:** As expected, the strength of the magnetic field increases proportionally with the current. The analytical results closely match the simulated values across all tested current levels, confirming the model’s ability to accurately capture this relationship.**Effect of Radius Variation:** Adjusting the coil radius results in changes to the magnetic field profile. In these cases, the analytical predictions continue to align well with the simulation data. This indicates that the model is capable of handling geometric changes effectively.

The comparison confirms that the proposed analytical approach performs well under both electrical and geometrical variations. It provides accurate estimations of magnetic field behavior, making it a practical and efficient tool for the design and optimization of planar coils in proximity sensing applications.

#### 3.4.1. Interpretation

Figure 7 demonstrates that the Simcenter MAGNET simulation results match the magnetic field calculations derived from the Biot–Savart (BS) law. The subplots visualize distinctive coil arrangements while varying parameters such as the current strength (I), coil radius (R), and number of coil turns (n). BS refers to the analytical calculations based on the Biot–Savart law, while FEM represents the Finite Element Method simulations performed using Simcenter MAGNET.

#### 3.4.2. Top Row of Figure 7: Current Variations

The magnetic field distributions in the top row display results for one turn and a fixed coil radius of 3 mm, implementing current levels from 1 A to 0.4 A.

The following remarks can be made:▪The magnetic field strength decreases proportionally with the applied current level.▪All calculated data matches closely with simulated data points, showing little deviation, under 2%, for each planar coil configuration.▪The field profile shows symmetrical distribution along the *z*-axis because the coil maintains uniform geometry, and it indicates that current directly affects B field strength.

#### 3.4.3. Bottom Row of Figure 7: Radius Variations

This row displays the field pattern measurements for coils with 1 A current flow and one turn while their radii vary between 2.5 mm and 1 mm.

The following remarks can be made:▪The magnetic field reaches its highest intensity level when the coil radius is smallest, because decreasing coil size creates a more focused field distribution.▪Magnetic field calculations based on the Biot–Savart law and simulations performed with Simcenter MAGNET match each other.▪Peak values obtained by calculations fall just above values determined through simulation.

### 3.5. Magnetic Field Distribution Using Analytical Model

The analytical model served to investigate the magnetic field distribution after the initial analysis. The data shows magnetic field strength (B) distribution through color-coded plots across the xy−plane at different *z*-axis positions from the planar coil (see Figure 8). The method generates thorough insights into magnetic field behavior, which both proves the validity of theoretical models and shows field attributes that extend past the coil area.

This study examines a single-turn planar coil with a 3.0 mm radius and 0.4 A current. The analytical model calculated magnetic field strength values at multiple points in the xy−plane while the observation plane traversed the *z*-axis from −5 mm to 5 mm with 0.001 mm increments, resulting in 10,000 magnetic field distribution images. The plots show magnetic field vector magnitudes at each data point to create complete field distribution maps for all planes. The method provides consistent and accurate evaluation of the spatial field distribution.

### 3.6. Enhanced Dataset Control and Validation

The magnetic field distributions show identical results to those generated by Simcenter MAGNET simulations. The essential characteristics include the field centralization and radial decay pattern and diffuse field distribution at extended distances, showing identical results between the analytical calculations and Simcenter MAGNET simulations. Therefore, in this study, the dataset is created by means of the analytical approach, i.e., the Biot–Savart law (see Equation (8)).

The analytical model enables dataset control with precision by allowing the modification of resolution settings and image count. The system provides valuable capabilities to create specialized datasets which match machine learning requirements, including those for training convolutional neural networks (CNNs).

In Figure 9a, a high-detail field distribution with 512×512 pixels at a 0.05 mm grid space is shown. This fine resolution enables accurate magnetic field evaluation throughout the whole plane, helping to obtain a detailed dataset suitable for neural network training purposes. The detailed field pattern features become accessible to the CNN through this high-resolution dataset, which promotes the better accuracy and performance of the model.

Figure 9b depicts the same magnetic field distribution shown in Figure 9a, but with a lower resolution (grid spacing of 0.2 mm). Reduced resolution enables faster data processing and smaller file sizes which are important for the analysis of magnetic field trends.

The compromised resolution in the image delivers a sufficient field structure to address training requirements and operational efficiency.

## 4. CNN-Based Analysis of Planar Coil Designs

Planar coil system optimization occurs through the merging of magnetic field analysis with convolutional neural networks (CNNs). A dataset of RGB images showing magnetic field distributions is created through the Biot–Savart law calculations explained in the previous section. The key design steps consist of

Designing a CNN architecture which specializes in predicting planar coil parameters from magnetic field distributions generated by the analytical model explained previously;Developing a dataset which will enhance the efficiency of future coil design procedures.

### 4.1. Datasets for Magnetic Field Distribution

The Biot–Savart law determines the magnetic field calculation by analyzing the coil geometry and current distribution in the initial step. Key parameters include
▪The coil design, consisting of the radius, number of turns, wire width, and wire spacing;▪The numerical experiment, using linear current variations from 0 mA to 1 A through 0.1 mA steps to generate 10,000 images for current prediction;▪10,000 magnetic field distribution images for diameter prediction;▪1000 images for predicting the number of turns using the allowable range from calculations of the feasibility of turns;▪100 images for predicting the number of layers, varying the number of layers from 1 to 100 layers;▪Spatial resolution: 0.2 grid within a 10  mm×10 mm region (air box).

Generating a database for training the four types of CNNs is time-consuming due to several factors, including generation, preprocessing, and annotation. Below is a breakdown of the database generation in Table 1.

### 4.2. Field Adaptation for CNN Training

For each configuration and current step,

▪A 2D scatter plot was generated with field strength mapped to an RGB colormap;▪A total of 10,000 images were generated for training by the analytical model;▪All images were treated for consistency and inserted in the CNN as an RGB matrix, as illustrated in Figure 10.

### 4.3. Convolutional Neural Network Architecture for Planar Coil Parameterization

A CNN uses the RGB dataset to train for magnetic field distribution prediction to solve the inverse problem. The network architecture targets image regression tasks by extracting spatial features which relate to magnetic field patterns.

The CNN architecture solves a regression problem that is based on images as model input. This type of model, with the chosen deep architecture and use of dropout and pooling, is capable of learning intricate representations from the input data.

The depth in the CNN is relevant to the size of training datasets. For current prediction with 10,000 images, a network with 12 layers would perform effectively as depicted in Table 2 (see CNN1 and CNN2). On the other hand, CNN3 for the prediction of the turn number needs a deep architecture, with a moderately deeper model and deeper layers. All networks benefit from depth, which enables both learning effectiveness and generalization while dropout controls overfitting.

### 4.4. CNN Architecture Design and Justification

For this section, CNN1 is taken as an example, and the same methodology applies to the rest of the CNN design procedure. In designing the convolutional neural network used for current prediction, the CNN was carefully designed to process RGB images of spatial magnetic field distribution. The dataset contains 10,000 such images of size 256×256×3 for the magnetic field profile in images given different current values. Constructing a CNN for the purpose of learning relevant spatial features and maintaining computational efficiency and overfitting was the goal. CNN1 and CNN2 have the same architecture, and the current dataset and the diameter dataset have the same size and are of the same nature; therefore, the choice of having the same CNN structure for both current and diameter prediction is reasonable. The only difference is in the Y input label vector which in CNN1 represents the current value (A), in CNN2 represents the diameter (mm), in CNN3 represents the number of turns, and in CNN4 represents the number of layers.

We visualized the feature maps after the first convolutional layer (Conv2D 32) in order to obtain some insight into learned features and to support the design choices. The activation maps from the 32 filters of this layer, given a representative magnetic field image, are shown in Figure 11.

As illustrated in Figure 11, each of the filters has been designed to pick up on a distinct spatial aspect of the magnetic field. Consequently, several filters have a strong response to circular or radial features (Filters 2, 4, 11), reflecting the characteristics of the magnetic fields caused by current-carrying conductors. Additionally, gradients and edges at high frequency (local variations) are captured (Filters 6, 18, 28). These observations prove that the use of 32 filters at this stage is sufficient, as activating them offers coverage of all low-level features. We designed a filter depth and architecture based on the activation maps through tailoring, using three convolutional blocks—a smaller convolution block to extract low spatial features, a medium block as a complement in the middle, and a bigger convolution block to extract high-level features. Global Average Pooling is used instead of flattening in order to reduce parameters and overcome overfitting.

Overall, the CNN architecture was not arbitrarily selected, but rather systematically constructed and validated both empirically and by using a visual assessment approach.

### 4.5. CNN Hyperparameter Tuning

#### 4.5.1. CNN1 Tuning for Current Prediction

To find the best hyperparameters for CNN1, a systematic grid search method was executed for current prediction modeling. The grid search constitutes a popular method that determines suitable hyperparameter combinations through systematic tests to arrive at the most effective setup. This study evaluated 720 different combinations of learning rate, epoch, batch size, and input image size values to select the best tuning of the CNN.

A systematic grid search was conducted, tuning the following:
Learning Rate $\left(l_{r}\right):$
$10^{-6},\;10^{-5},\;10^{-4},\;10^{-3},\;10^{-2},10^{-1}$.Epochs: 10, 20, 30, 40, 50, 75.Batch Sizes: 2, 4, 8, 16, 32.Input image size: (64×64), (128×128), (256×256), and (512×512).

All results from the grid search testing of CNN1 are presented in Table 3. Performance indicators are MAE, MSE, and training time.

The analyzed grid showed particular combinations that exceeded others in terms of performance which helped identify how the model responded to hyperparameter adjustments. An evaluation of the results enabled the selection of CNN1’s optimal configuration which achieved a balance between accuracy outcomes and training performance.

#### 4.5.2. Evaluation Metrics

▪Mean Absolute Error (MAE): Average absolute error between predicted and true current values (mA).▪Mean Squared Error (MSE): Average squared current error (mA).

#### 4.5.3. Mean Absolute Error (MAE) Across Tuning Combinations

▪The visual representation demonstrates how Mean Absolute Error changes when using different parameter settings.▪The plot distinguishes learning rates through different shades and marker styles, which demonstrates that the lower learning rates $10^{-6}$ and $10^{-5}$ produce superior MAE values.

#### 4.5.4. Loss Across Tuning Combinations

▪Figure 12 depicts the relationship between the loss and tuning combinations, grouped by batch size.▪Smaller batch sizes generally achieve lower loss, while larger batch sizes demonstrate higher loss due to potential underfitting.

#### 4.5.5. MAE Across Epochs for Different Batch Sizes

Figure 13 depicts the evolution of MAE across different batch sizes along with rising epochs. The results show that smaller batch sizes continue to improve their performance with each epoch but larger batch sizes achieve stability sooner to attain accuracy and efficiency performance levels.

#### 4.5.6. Interpretation

The lines in the graph represent the mean MAE for each batch size as epochs increase. The gray shaded area represents a confidence interval that indicates the stability of the model’s performance.

Narrow shaded areas suggest high confidence and low variability, meaning that the results are stable and reproducible for that batch size. In contrast, wider areas indicate greater variability, highlighting potential instability in learning.

A grid search was implemented for tuning procedures for every CNN model, with their performance results displayed through the subsequent figures. A comprehensive model performance assessment includes the training curve, validation curve, and loss curves plotted next to the predicted vs. actual current values. The set of graphs below in Figure 14 illustrates both training and validation curves together with predicted vs. actual values.

### 4.6. Performance Comparison with Pre-Trained Architectures

To assess the effectiveness of the proposed convolutional neural network (CNN1), as a case study, a comparative analysis was conducted against two widely recognized architectures: ResNet18 and MobileNetV2. These models are frequently employed as benchmarks due to their demonstrated success in image classification tasks and their efficient design for feature extraction.

Both ResNet18 and MobileNetV2 were originally developed and pre-trained on the ImageNet dataset, which consists of millions of natural images across numerous object categories [17,18,19]. While their architecture is optimized for classification problems, the convolutional backbone of these models enables their adaptation for regression tasks by modifying the final layer to output continuous values rather than discrete classes. This adaptability allows them to be applied beyond classification to domains such as medical imaging, environmental monitoring, and, in this case, magnetic field distribution estimation.

The proposed CNN1, specifically designed and optimized for the domain-specific spatial characteristics of magnetic field maps, achieved superior performance in terms of mean squared error (MSE) and Mean Absolute Error (MAE) when compared to the adapted ResNet18 and MobileNetV2 models. Notably, CNN1 attained these results with significantly fewer parameters, highlighting the efficiency of a tailored architecture over general-purpose networks.

Table 4 summarizes the training hyperparameters and performance metrics of the three models.

Based on Table 4, the proposed CNN1 outperforms both ResNet18 and MobileNetV2 in terms of mean squared error (MSE) and Mean Absolute Error (MAE), despite having a significantly smaller number of parameters. This demonstrates that carefully designing a neural network tailored to the specific domain and the unique characteristics of the dataset can lead to substantial improvements in predictive performance when compared to employing general-purpose, pre-trained models originally developed for natural images.

The distinct nature of magnetic field distribution data, with its domain-specific spatial and physical features, calls for customized model architectures that can better capture relevant patterns and relationships. By focusing on the data’s intrinsic properties during model development, it is possible to achieve enhanced accuracy and generalization.

Furthermore, the reduced model complexity and parameter count of CNN1 translate into lower computational and memory requirements, which is a critical advantage for real-world applications where resource constraints often limit the feasibility of deploying large-scale models. This efficiency not only facilitates faster training and inference times but also enables potential implementation on embedded systems or edge devices commonly used in sensor-based technologies.

In summary, the results advocate for a paradigm shift from relying solely on off-the-shelf deep learning models toward the development of bespoke architectures that align closely with the underlying physics and data characteristics of specialized domains.

## 5. Application Example: Coil Geometry Design from Magnetic Field Distribution

The trained CNNs are now tested for solving the inverse problem, starting from a prescribed magnetic field distribution without any prior knowledge about the geometry of the planar coil that generated the illustrated magnetic field distribution (see Figure 15a).

To test how well the proposed CNN-based model works, we used an image of magnetic fields as input. This image shows the strength of the magnetic field in 2D, where the color intensity of each pixel represents the value of the magnetic field at that point.

The model takes an input image with a size of 256×256 pixels. The image does not reflect a real measured or simulated magnetic field distribution, but a distribution drawn by hand (prescribed magnetic field distribution by a designer) to assess the results. After the image is given to the CNN, it predicts the shape and size of a planar coil that would produce a similar magnetic field.

Figure 15 depicts the multiple stages of creating a planar coil using the developed model, assuming that we want to obtain a magnetic field strength of 0.4 mT at a 3.5 mm radius, 0.6 mT at 2.5 mm, 1 mT at 2 mm, 1.3 mT at 1.7 mm, and finally 2 mT in the center of the coil up to a 1 mm radial distance. Of course, the chosen values probably will not respect the physical laws that govern the distribution of the magnetic field; nevertheless, we want to obtain a planar coil that would generate a magnetic field close to our requirements.

The target magnetic field is shown in Figure 15a, which was manually created using a drawing tool. The input image has a size of 256×256, as this resolution was previously found to give the best results (see Table 5).

The predicted parameters include the current, diameter, number of turns, and number of layers. Subsequently the fill ratio is calculated from the predicted parameters along with the overall size of the planar coil.

To evaluate accuracy, the magnetic field produced by the generated coil is compared to the input magnetic field.

The model for the given input data identifies an optimal coil setup as shown below.

This test demonstrates that our method is practical and can be used to design planar coils efficiently, without requiring multiple design steps. Figure 15b shows the generated planar coil in 2D, while Figure 15c presents its 3D structure. In Figure 15d, the produced magnetic field is displayed, and it visually matches the input magnetic field. These results suggest that the model can be applied in real-world scenarios, such as inductive sensors, wireless power transfer systems, and electromagnetic design software.

In addition to visual comparison, we used quantitative metrics to evaluate how closely the generated magnetic field matched the target field. Specifically, we calculated the mean squared error (MSE) across the 256×256 grid. In most cases, the MSE was below $0.18\;mT^{2}$, which indicates a strong match between the generated and input magnetic fields. This shows that the model not only produces visually similar results but also maintains a high level of numerical accuracy, making it suitable for real-world applications.

Figure 16 demonstrates an experiment with a smoothed curve, generated using interpolation from magnetic field data extracted from the input image. The curve of the output magnetic field corresponds to the magnetic field of the planar coil generated by the model.

The mean squared error (MSE), found to be $0.179\;mT^{2}$, corresponds to an average error of approximately 0.424 mT (RMSE) and average deviation (MAE) of 0.353 mT. This minor error in the results is mainly due to the magnetic field image distribution for the test being determined manually, which does not fully show coherence with the principles of electromagnetic wave propagation.

The model also performed consistently across various test cases. When different magnetic field patterns were used as input, the neural network was able to generate suitable coil parameters without requiring manual adjustments. This suggests that the model can adapt to a wide range of design needs and is not limited to a specific type of input.

## 6. Conclusions

This research produced an adaptable structure which combines electromagnetic modeling and deep learning methods to optimize planar coil performance.

The hybrid modeling method utilizes electromagnetic principles with machine learning algorithms to enhance traditional coil simulation efficiency by effectively managing computational problems.
▪The coil geometry prediction model was very accurate, with a less than 2% difference compared to Simcenter MAGNET simulations for 50 different coil setups. It studied currents from 0.4 A to 1 A, radii from 2 mm to 3 mm, and various layer configurations, accurately representing both simulated and theoretical coil designs while reliably detecting magnetic fields.▪A dataset of 62,200 RGB images showing magnetic field patterns was used to train a CNN, which achieved a very low mean squared error (less than 0.01) during validation. This allowed fast and precise magnetic field predictions without the running of slow simulations.▪The best results were achieved when using low learning rates (between $10^{6}$ and $10^{5}$) and small batch sizes (two or four), which reduced test errors and improved model generalization. High learning rates and large batch sizes caused poor training and overfitting, increasing errors.▪This study showed that magnetic field strength depends on coil size and current. For example, coils with a 2 mm radius and 1 A current produced 50% stronger peak fields than coils with a 3 mm radius at the same current. This is important for applications like inductive sensing or wireless power transfer.▪Processing time grows with dataset size and image resolution. For example, it increased from about 5.5 h to nearly 28 h when image size went from 64×64  to 256×256 for 10,000 images. The total processing time for all datasets was estimated at over 98 h.▪The model learned practical design rules on its own, avoiding coil shapes that are hard to build or too large. These rules are usually added manually in traditional methods, so this makes design easier.▪The method is very fast, taking less than one second on a normal GPU to predict and generate coil designs. This speed makes it useful for interactive tools where quick results are needed, much faster than traditional simulations or optimizations.▪The model can convert a magnetic field image into a workable coil design that keeps the field’s shape and strength. The generated coils are compact, are smooth, and have no sharp corners, which helps reduce energy loss and makes manufacturing easier—key for sensors or wireless charging systems.▪This method can be part of bigger design systems, for example, inside optimization loops where many coil designs need to be tested quickly.▪The method represents a new way to combine machine learning with electromagnetism by learning from data while following magnetic rules. It could be combined with traditional tools to make coil design faster and more flexible, especially when balancing multiple goals related to field shape, heat, or size.▪Future improvements could include letting users set specific goals, like optimizing for certain frequencies or avoiding interference with multiple coils, and adding smart learning or multi-goal optimization to make the tool even more helpful for fast and flexible design.

Overall, this study highlights how combining machine learning with classical modeling can create efficient, precise, and adaptable electromagnetic designs useful across engineering and science.

## Figures and Tables

**Figure 1 sensors-25-04429-f001:**
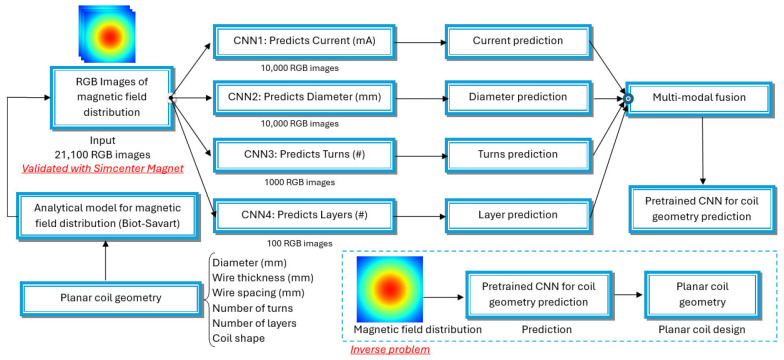
Workflow for planar coil geometry design and inverse problem solving.

**Figure 2 sensors-25-04429-f002:**
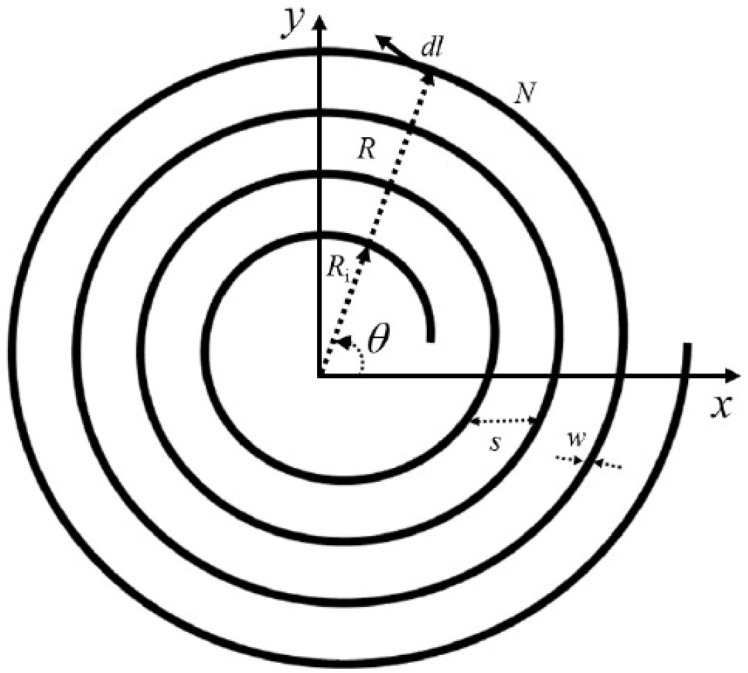
Spiral planar coil layout.

**Figure 3 sensors-25-04429-f003:**
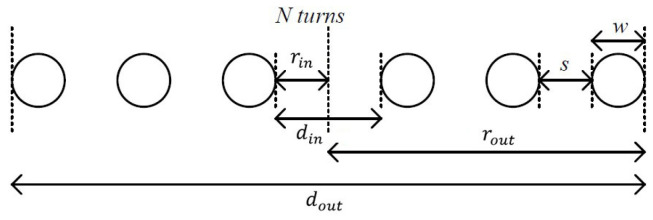
Cross-sectional view of planar spiral coil.

**Figure 4 sensors-25-04429-f004:**
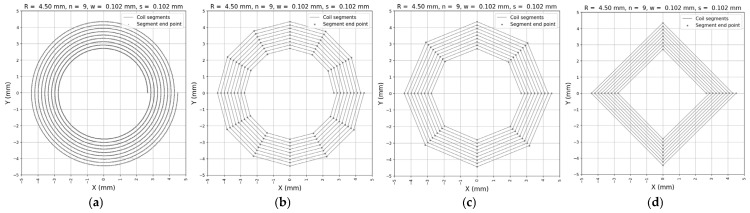
Comparison of spiral coil discretization at different angular resolutions: (**a**) 1 degree per coil point; (**b**) 30 degrees per coil point; (**c**) 45 degrees per coil point; (**d**) 90 degrees per coil point (square-shape planar coil).

**Figure 5 sensors-25-04429-f005:**
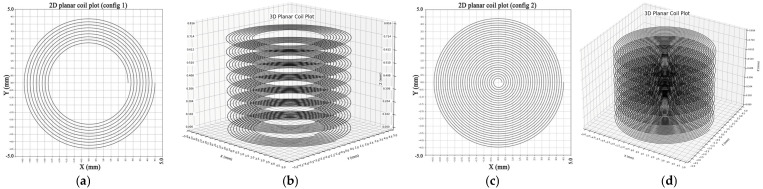
Example of planar coil designs generated by the analytical model: (**a**) 2D planar coil design, config 1; (**b**) 3D stacked planar coil, config 1; (**c**) 2D planar coil design, config 2; (**d**) 3D stacked planar coil, config 2.

**Figure 6 sensors-25-04429-f006:**
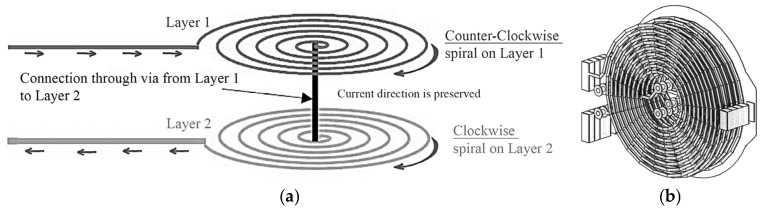
Representation of a planar coil with current flow and axial orientation for Simcenter MAGNET simulation. (**a**) Stacked overview of the designed planar coil; (**b**) real 3D implementation of the planar coil.

**Figure 7 sensors-25-04429-f007:**
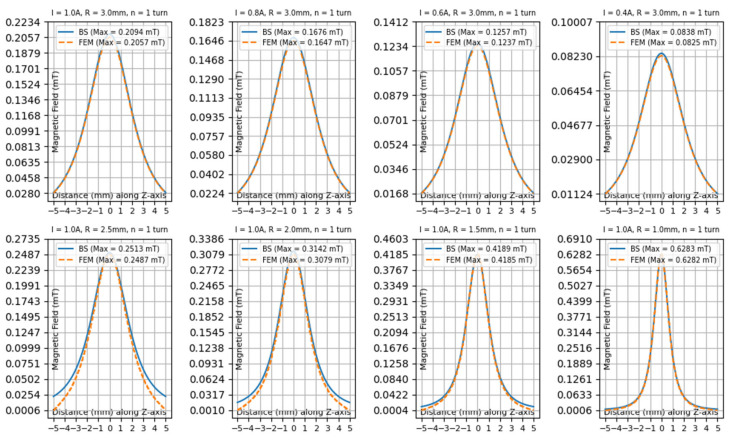
Comparison of magnetic field calculations using Biot–Savart model (BS) and Simcenter MAGNET simulations (FEM) for various coil configurations.

**Figure 8 sensors-25-04429-f008:**
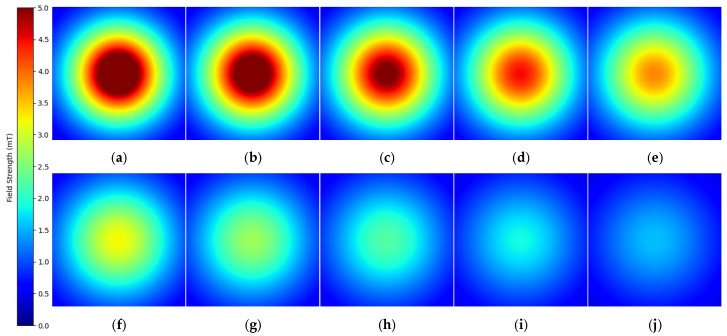
Evolution of magnetic field distribution (B) along the *z*-axis generated by a circular planar coil: (**a**) magnetic field distribution (B) at z = 0 mm; (**b**) z = 0.5 mm; (**c**) z = 1.0 mm; (**d**) z = 1.5 mm; (**e**) z = 2.0 mm; (**f**) z = 2.5 mm; (**g**) z = 3.0 mm; (**h**) z = 3.5 mm; (**i**) z = 4.0 mm; (**j**) z = 4.5 mm.

**Figure 9 sensors-25-04429-f009:**
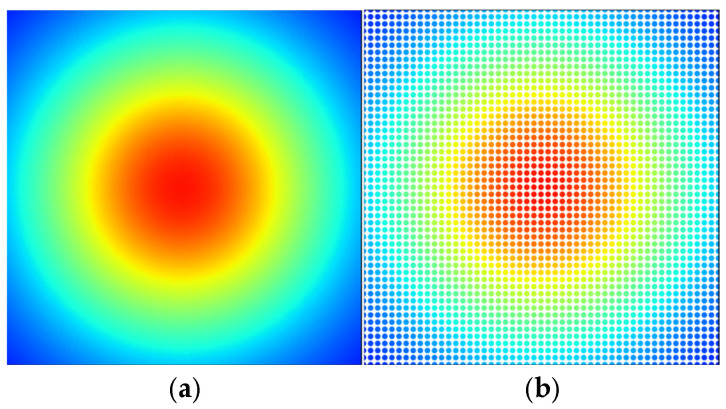
Dataset adjustment of the magnetic field distribution: high resolution and coarse grid. (**a**) High-resolution magnetic field distribution, 512×512 px, resolution=0.05 mm; (**b**) low-resolution magnetic field distribution, 512×512 px, resolution=0.2 mm.

**Figure 10 sensors-25-04429-f010:**
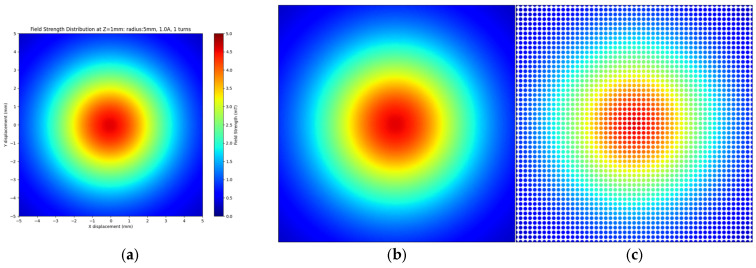
Dataset visualization; (**a**) magnetic field distribution RGB image; (**b**) cropped magnetic field distribution RGB image; (**c**) cropped and downscaled magnetic field distribution RGB image.

**Figure 11 sensors-25-04429-f011:**
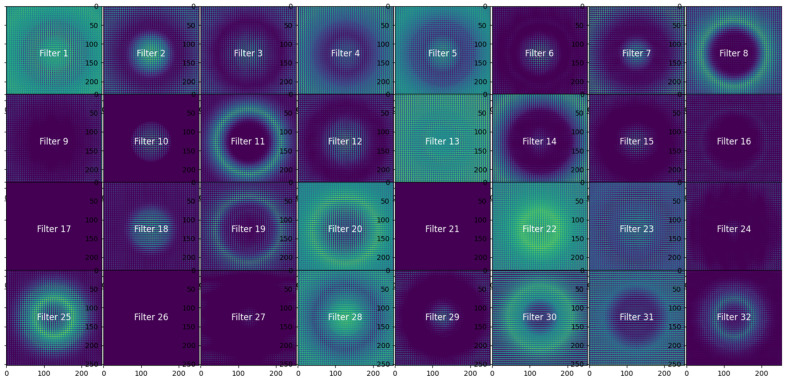
Feature maps at CNN1 Layer 3 (Conv2D-32), showing 32 distinct filter responses to a single input image.

**Figure 12 sensors-25-04429-f012:**
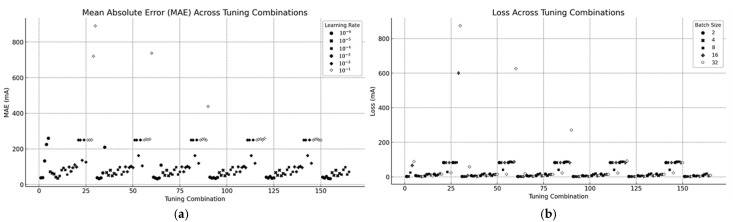
Results of tuning combinations: (**a**) MAE across tuning combinations; (**b**) loss variation across tuning combinations grouped by batch size.

**Figure 13 sensors-25-04429-f013:**
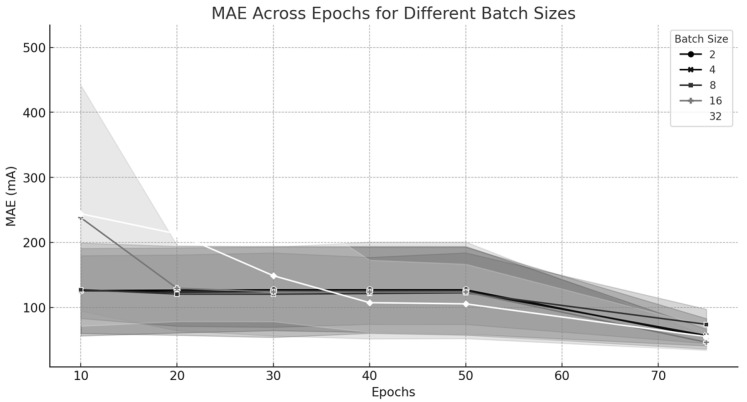
MAE progression over epochs for various batch sizes.

**Figure 14 sensors-25-04429-f014:**
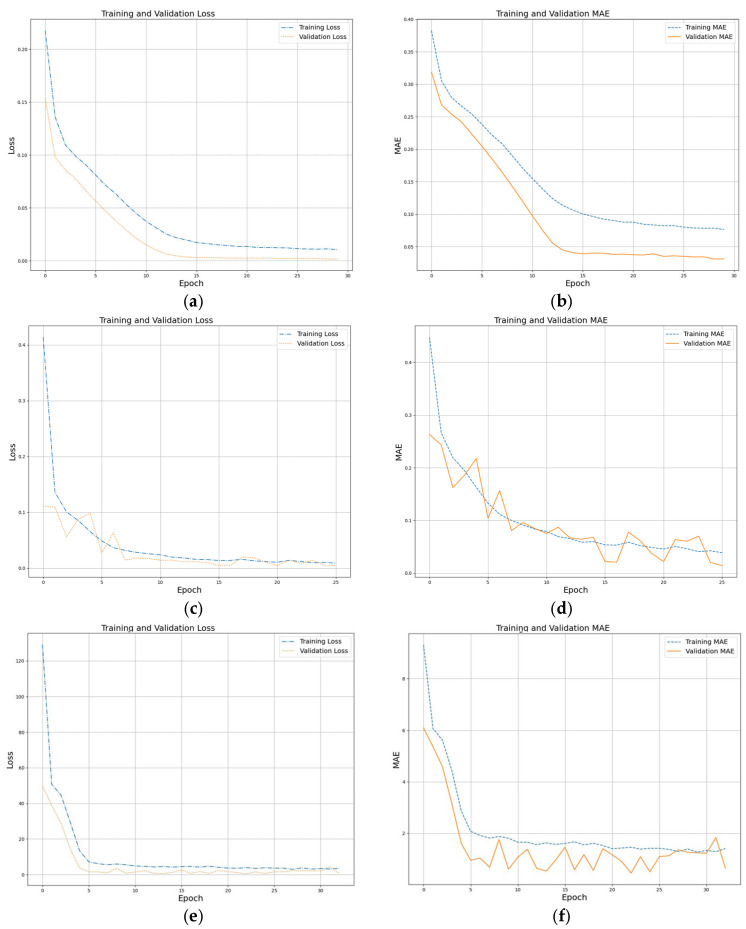
Training curves for best configurations across all CNNs: (**a**) training and validation loss for current prediction; (**b**) training and validation MAE for current prediction; (**c**) training and validation loss for diameter prediction; (**d**) training and validation MAE for diameter prediction; (**e**) training and validation loss for the prediction of the number of turns; (**f**) training and validation MAE for the prediction of the number of turns; (**g**) training and validation loss for the prediction of the number of layers; (**h**) training and validation MAE for the prediction of the number of layers.

**Figure 15 sensors-25-04429-f015:**
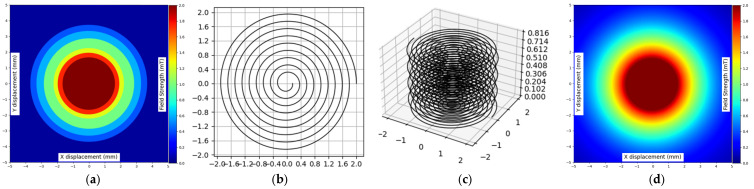
Prediction of planar coil geometry from magnetic field distribution: (**a**) random magnetic field distribution input drawn manually; (**b**) predicted planar coil geometry in 2D; (**c**) predicted planar coil geometry in 3D; (**d**) the predicted magnetic field distribution plotted by the model.

**Figure 16 sensors-25-04429-f016:**
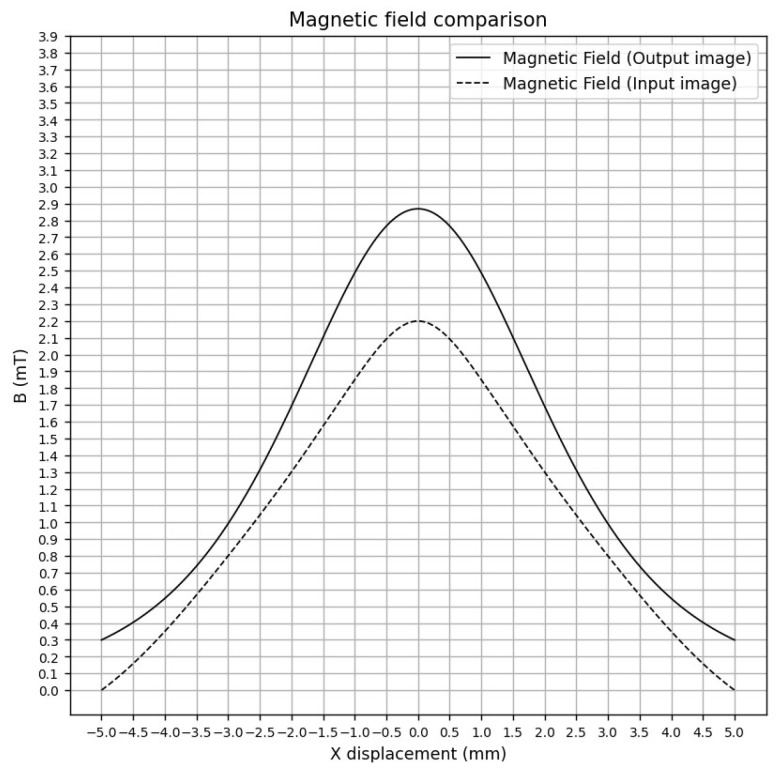
Magnetic field comparison between the magnetic field input and output of the model.

**Table 1 sensors-25-04429-t001:** Estimated time for CNN dataset generation based on image size and dataset type.

Dataset	Total Number of Images	Estimated Time per Image	Image Size	Total Time
Current	10,000	2 s	64×64	05:33:20
	10,000	5 s	128×128	13:53:20
	10,000	10 s	256×256	27:46:40
Diameter	10,000	2 s	64×64	05:33:20
	10,000	5 s	128×128	13:53:20
	10,000	10 s	256×256	27:46:40
Turns	1000	2 s	64×64	00:33:20
	1000	5 s	128×128	01:23:20
	1000	10 s	256×256	02:46:40
Layers	100	2 s	64×64	00:03:20
	100	5 s	128×128	00:08:20
	100	10 s	256×256	00:16:40
Total	62,200	-	-	98:24:00

**Table 2 sensors-25-04429-t002:** Architecture overview of CNN models for the multiple regression tasks.

CNN	Layer Name	Output Shape	№ Parameters	Activation Function
CNN1 and CNN2 Current and Diameter Prediction	Layer 1: Input	(None, 256, 256, 3)	0	N/A
	Layer 2: Rescaling	(None, 256, 256, 3)	0	N/A
	Layer 3: Conv2D-32	(None, 254, 254, 32)	896	ReLU
	Layer 4: MaxPool-2 × 2	(None, 127, 127, 32)	0	N/A
	Layer 5: Conv2D-64	(None, 125, 125, 64)	18,496	ReLU
	Layer 6: MaxPool-2 × 2	(None, 62, 62, 64)	0	N/A
	Layer 7: Conv2D-128	(None, 60, 60, 128)	73,856	ReLU
	Layer 8: MaxPool-2 × 2	(None, 30, 30, 128)	0	N/A
	Layer 9: GlobalAvgPool	(None, 128)	0	N/A
	Layer 10: Dense-128	(None, 128)	16,512	ReLU
	Layer 11: Dropout	(None, 128)	0	N/A
	Layer 12: Output	(None, 1)	129	ReLU
CNN3 Prediction of Number of Turns	Layer 1: Input	(None, 256, 256, 3)	0	N/A
	Layer 2: Rescaling	(None, 256, 256, 3)	0	N/A
	Layer 3: Conv2D-32	(None, 256, 256, 32)	896	ReLU
	Layer 4: BatchNorm	(None, 256, 256, 32)	128	N/A
	Layer 5: Conv2D-32	(None, 256, 256, 32)	9248	ReLU
	Layer 6: MaxPool	(None, 128, 128, 32)	0	N/A
	Layer 7: Dropout	(None, 128, 128, 32)	0	N/A
	Layer 8: Conv2D-64	(None, 128, 128, 64)	18,496	ReLU
	Layer 9: BatchNorm	(None, 128, 128, 64)	256	N/A
	Layer 10: Conv2D-64	(None, 128, 128, 64)	36,928	ReLU
	Layer 11: MaxPool	(None, 64, 64, 64)	0	N/A
	Layer 12: Dropout	(None, 64, 64, 64)	0	N/A
	Layer 13: Conv2D-128	(None, 64, 64, 128)	73,856	ReLU
	Layer 14: BatchNorm	(None, 64, 64, 128)	512	N/A
	Layer 15: Conv2D-128	(None, 64, 64, 128)	147,584	ReLU
	Layer 16: MaxPool	(None, 32, 32, 128)	0	N/A
	Layer 17: Dropout	(None, 32, 32, 128)	0	N/A
	Layer 18: GlobalAvgPool	(None, 128)	0	N/A
	Layer 19: Dense-128	(None, 128)	16,512	ReLU
	Layer 20: Dropout	(None, 128)	0	N/A
	Layer 21: Dense-64	(None, 64)	8256	ReLU
	Layer 22: Output	(None, 1)	65	N/A
CNN4 Prediction of Number of Layers	Layer 1: Input	(None, 256, 256, 3)	0	N/A
	Layer 2: Rescaling	(None, 256, 256, 3)	0	N/A
	Layer 3: Conv2D-64	(None, 256, 256, 64)	1792	ReLU
	Layer 4: BatchNorm	(None, 256, 256, 64)	256	N/A
	Layer 5: Conv2D-64	(None, 256, 256, 64)	36,864	ReLU
	Layer 6: MaxPooling	(None, 128, 128, 64)	0	N/A
	Layer 7: Dropout	(None, 128, 128, 64)	0	N/A
	Layer 8: Conv2D-128	(None, 128, 128, 128)	73,856	ReLU
	Layer 9: BatchNorm	(None, 128, 128, 128)	512	N/A
	Layer 10: Conv2D-128	(None, 128, 128, 128)	147,584	ReLU
	Layer 11: MaxPooling	(None, 64, 64, 128)	0	N/A
	Layer 12: Dropout	(None, 64, 64, 128)	0	N/A
	Layer 13: Conv2D-256	(None, 64, 64, 256)	295,168	ReLU
	Layer 14: BatchNorm	(None, 64, 64, 256)	1024	N/A
	Layer 15: Conv2D-256	(None, 64, 64, 256)	590,080	ReLU
	Layer 16: MaxPooling	(None, 32, 32, 256)	0	N/A
	Layer 17: Dropout	(None, 32, 32, 256)	0	N/A
	Layer 18: GlobalAvgPool	(None, 256)	0	N/A
	Layer 19: Dense-256	(None, 256)	65,792	ReLU
	Layer 20: Dropout	(None, 256)	0	N/A
	Layer 21: Dense-128	(None, 128)	32,896	ReLU
	Layer 22: Output	(None, 1)	129	N/A

**Table 3 sensors-25-04429-t003:** Hyperparameter tuning results for CNN1 (current prediction). Best config is highlighted.

Config	Learning Rate	Epochs	Batch Size	MAE (mA)	MSE (mA)	Training Time	Image Size
55	1.00 × 10^−2^	20	32	104.71	15.61	00:11:40	256×256
56	1.00 × 10^−1^	20	2	249.43	82.65	00:08:58	256×256
57	1.00 × 10^−1^	20	4	253.61	86.92	00:10:13	256×256
58	1.00 × 10^−1^	20	8	251.47	84.71	00:11:48	256×256
59	1.00 × 10^−1^	20	16	254.1	87.45	00:11:43	256×256
60	1.00 × 10^−1^	20	32	736.93	625.5	00:11:35	256×256
61	1.00 × 10^−6^	30	2	40.86	2.143	00:15:28	256×256
62	1.00 × 10^−6^	30	4	36.05	1.886	00:16:26	256×256
63	1.00 × 10^−6^	30	8	30.98	1.57	00:17:47	256×256
64	1.00 × 10^−6^	30	16	36.49	1.929	00:17:44	256×256
65	1.00 × 10^−6^	30	32	108.32	18.13	00:17:33	256×256
66	1.00 × 10^−5^	30	2	66.73	6.952	00:07:57	256×256
67	1.00 × 10^−5^	30	4	51.45	4.23	00:09:07	256×256
68	1.00 × 10^−5^	30	8	79.91	7.598	00:07:45	256×256
69	1.00 × 10^−5^	30	16	48.78	2.975	00:09:33	256×256
70	1.00 × 10^−5^	30	32	61.48	4.297	00:14:36	256×256
71	1.00 × 10^−4^	30	2	56.07	6.865	00:08:31	256×256
72	1.00 × 10^−4^	30	4	82.85	14.38	00:06:43	256×256
73	1.00 × 10^−4^	30	8	97.2	19.37	00:06:35	256×256
74	1.00 × 10^−4^	30	16	56.33	6.248	00:09:31	256×256
75	1.00 × 10^−4^	30	32	71.29	8.507	00:07:40	256×256
76	1.00 × 10^−3^	30	2	99.52	16.16	00:07:44	256×256
77	1.00 × 10^−3^	30	4	71.8	7.48	00:06:45	256×256
78	1.00 × 10^−3^	30	8	95.71	14.59	00:06:33	256×256
79	1.00 × 10^−3^	30	16	102.56	15.08	00:06:35	256×256
80	1.00 × 10^−3^	30	32	94.72	14.53	00:06:30	256×256
81	1.00 × 10^−2^	30	2	249.2	82.46	00:07:43	256×256
82	1.00 × 10^−2^	30	4	249.19	82.44	00:07:15	256×256
83	1.00 × 10^−2^	30	8	161.93	41.15	00:09:24	256×256
84	1.00 × 10^−2^	30	16	249.59	82.87	00:06:31	256×256
85	1.00 × 10^−2^	30	32	118.79	22.44	00:13:57	256×256
88	1.00 × 10^−1^	30	8	255.39	88.44	00:17:28	256×256

**Table 4 sensors-25-04429-t004:** Comparison of model complexity and regression performance.

Model	Parameters	Learning Rate	Epochs	Batch Size	MAE (mA)	MSE (mA)
CNN 1	118,081	$10^{-6}$	30	8	30.98	1.57
ResNet18	11,689,512	$10^{-3}$	30	4	34.10	1.6
MobileNetV2	3,504,872	$10^{-2}$	20	32	45.08	3.96

**Table 5 sensors-25-04429-t005:** The designed planar coil geometry for the given input data.

Parameter	Diameter	Current	Turns	Layers	w(mm)	s(mm)	ls(mm)	Coil Thickness
Value	4 mm	54 mA	9	8	4 mils	4 mils	4 mils	0.714 mm

## Data Availability

The original contributions presented in this study are included in the article.

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
