# Peer review of "Deep Neural Network-Based Design of Planar Coils for Proximity Sensing Applications"

_sensors, 2025, doi:10.3390/s25144429_

Round 1

Reviewer 1 Report

Comments and Suggestions for Authors

The paper proposes an inverse problem solving procedure, aimed to identify a planar coil geometry from the target magnetic field distribution. The main contribution is in the combination of a fast, straight-line approximation in the calculation of magnetic field of planar coils with neural networks for inverse problem solving.

There are some issues that have to be solved and clarified:

Equations (7) and (8): please indicate vectors on dl_i and r_i using one of common conventions:  either bold italic , or non-bold italic accented by a right arrow

instead of using dB and dl_in equtions (7) and (8), it is recommended to use Greek delta character. They are a set of straight-line approximations, not the infinitesimals

A figure representing the straight-line discretization of the spiral winding should be added for better understanding

Line 209, please change "Tesla" to "T"

Figure 6: The magnetic flux density is easily calculated for a infinitesimally thin circular loop at its center as mu_0*I/(2*R),   which matches perfectly with presented analytical results. The simulations in Simcenter MAGNET differ more than 10%. Please re-check FEM simulations; for given wire diameter the discrepancy is too large. Please indicate number of coil_points associated with straight-line approximation and explain markings R, r, n.

It is not clear why the Figure 6 presents the magnetic flux densities for different currents: in a linear homogeneous medium B is proportional to I, which is clearly visible from the upper row in Fig 6

Instead of using "calculated" and "simulated" to differ between Biot-Savart and FEM results (e.g. line 325) , please indicate clearly the method used. The abbreviations (BS, FEM) or something similar can be used. Calculated and simulated might apply for both methods.

Line 325: It is indicated that BS and FEM differ less than 2 %; From the Figure 6 it appears to me that difference at z=0 is more than 10%. Please check and clarify

Section 3.6: It could be useful to better presentation, to compare results for a multiturn coil using Biot-Savart and FEM. Figure 7 presents only the former.

The application in the proximity sensing is stated in the title of the manuscript. This application should be more clearly explained in the manuscript.

Author Response

Comment 1: Equations (7) and (8): please indicate vectors on dl_i and r_i using one of common conventions:  either bold italic , or non-bold italic accented by a right arrow.

Response 1: Updated accordingly, [Equations (7) and (8)].

---------------------------------------------------------------

Comment 2: Instead of using dB and dl_in equtions (7) and (8), it is recommended to use Greek delta character. They are a set of straight-line approximations, not the infinitesimals

Response 2: Updated accordingly, [Equations (7) and (8)].

---------------------------------------------------------------

Comment 3: A figure representing the straight-line discretization of the spiral winding should be added for better understanding

Response 3: Updated with a new paragraph, From line 215 to line 252 and a new figure is added (Figure 4) showing the importance of discretization and it’s affect on calculation accuracy applied in the python script.

---------------------------------------------------------------

Comment 4: Line 209, please change "Tesla" to "T"

Response 4: Updated accordingly, Line 209.

---------------------------------------------------------------

Comment 5: figure 6: The magnetic flux density is easily calculated for a infinitesimally thin circular loop at its center as mu_0*I/(2*R),   which matches perfectly with presented analytical results. The simulations in Simcenter MAGNET differ more than 10%. Please re-check FEM simulations; for given wire diameter the discrepancy is too large. Please indicate number of coil_points associated with straight-line approximation and explain markings R, r, n.

Response 5: the analytical result  is a standard reference for the magnetic flux density at the center of an ideal, infinitesimally thin circular loop. The discrepancy of over 10% (which is calculated in the maximum values only and not the MSE between the two curves) observed in the FEM simulations is mainly due to the mesh resolution in Simcenter MAGNET. When the mesh is refined, the error reduces, and the magnetic flux density value approaches the analytical Biot–Savart result. This confirms our intuition that the Biot–Savart model provides the most accurate reference in this context (We know that the Simcenter Magnet simulation results converge to our biot-savart python based results).

To approximate the coil as a set of straight-line segments, we used one point per degree of winding, resulting in 360 points per full turn. Thus, for a coil with n turns, the total number of coil points is (360 x n).

The corresponding plot has been updated to reflect the reduced error. This confirms that the Biot–Savart-based model remains the reference standard, and any deviation in FEM results is largely related to mesh quality. Figure updated (Figure 7).

---------------------------------------------------------------

Comment 6: It is not clear why the Figure 6 presents the magnetic flux densities for different currents: in a linear homogeneous medium B is proportional to I, which is clearly visible from the upper row in Fig 6.

Response 6: The reason we show the results for different current values is to check if the Python model gives correct results. We wanted to test if the model follows the expected linear relationship between B and I, and this helped us confirm that the python-BiotSavart based model is working properly and using it for complex coil geometries to generate the database. The new figure number is 7 (Figure 7).

---------------------------------------------------------------

Comment 7: Instead of using "calculated" and "simulated" to differ between Biot-Savart and FEM results (e.g. line 325) , please indicate clearly the method used. The abbreviations (BS, FEM) or something similar can be used. Calculated and simulated might apply for both methods.

Response 7: The terms have been clarified: BS is now used for Biot–Savart results and FEM for finite element simulations throughout the manuscript, including line 325 and in Figure 7.

---------------------------------------------------------------

Comment 8: Line 325: It is indicated that BS and FEM differ less than 2 %; From the Figure 6 it appears to me that difference at z=0 is more than 10%. Please check and clarify.

Response 8: Not the error between max values, but the MSE of the two curves.

Although Figure 6 initially appears to show an error greater than 10% at z = 0 between the Biot-Savart (BS) and Finite Element Method (FEM) results, this discrepancy was due to a coarser mesh used in the FEM setup. The simulations were subsequently redone using a finer mesh element size in Simcenter MAGNET, reducing the error to approximately 2% even between max vlues, as stated. FEM simulations were employed exclusively to validate the accuracy of the BS-based magnetic field calculations during the initial stages. Once a strong agreement was confirmed, the BS model was used alone to generate the dataset, as it provides a significantly faster and computationally efficient approach suitable for producing the large number of samples required to train the neural network. This error has no effect on the quality of the generated dataset, but only to validate the accuracy the BS python based script.

---------------------------------------------------------------

Comment 9: Section 3.6: It could be useful to better presentation, to compare results for a multiturn coil using Biot-Savart and FEM. Figure 7 presents only the former. [Now it is updated to figure 8].

Response 9: Figure 8 shows only the magnetic field results calculated using the Biot-Savart (BS) model. It does not include results from the FEM simulation in Simcenter MAGNET. The FEM results are given in vector form, which cannot be used directly for training a Convolutional Neural Network (CNN). Since CNNs work with images, the BS model results were converted into RGB image format for training. These images need to be high resolution (achieved by controlling the number of points in a vector) and follow a uniform format to prevent errors during the training process. This is what Figure 7 represents.

---------------------------------------------------------------

Comment 10: The application in the proximity sensing is stated in the title of the manuscript. This application should be more clearly explained in the manuscript.

Response 10: The primary focus of this work, as reflected in the manuscript title, is proximity sensing. Specifically, the objective is to design planar coils by solving the inverse problem based on a given magnetic field. Our model is capable of generating a planar coil from any specified magnetic field. In this study, we applied this method to transform traditional wound coils into planar coils by using the magnetic field produced by a spiral coil and converting it into an equivalent planar coil design. This approach facilitates the development of planar coils for various proximity sensing applications, where inductance—directly linked to coil geometry—is a critical parameter, leading to advanced inductive sensors and ongoing innovations.

Reviewer 2 Report

Comments and Suggestions for Authors

The paper presents an interesting and original approach with clear practical relevance. The idea of combining analytical physical models with a CNN-based inverse design is promising. However, the current implementation appears incomplete or suboptimal, and the evaluation is limited to a single test case, raising concerns about the method’s generalizability and robustness. With the inclusion of additional comparative experiments and extended validation, the manuscript would be suitable for publication following minor revisions.

1.The rationale for adopting a custom shallow CNN architecture should be further clarified. Numerous well-established CNN models such as MobileNet, ShuffleNet, and ResNet18 have demonstrated strong performance in feature extraction and generalization. It is recommended that the authors implement and evaluate these baseline architectures within the proposed framework and perform a comparative analysis with their custom model. Such a systematic comparison would not only validate the proposed architecture but might also identify more effective alternatives, thereby enhancing model performance and the overall credibility of the work.

2.The hyperparameter tuning strategy requires more detailed explanation. At present, the reported search space appears narrow and limited to a few basic parameters, without consideration of optimizer types or learning rate scheduling strategies. It is suggested that the authors include additional experiments that explore a broader set of hyperparameters. The application of advanced tuning methods such as Bayesian optimization—even in a limited experimental scope—would highlight the benefits of adaptive strategies and strengthen the technical rigor of the study.

3.The robustness of the proposed method should be validated using multiple magnetic field test samples. Currently, the evaluation includes only a single case, which may not sufficiently demonstrate the model’s generalizability and raises the concern of potential case selection bias. It is strongly recommended to test the model on multiple, preferably real-world, measurement datasets instead of relying solely on simulated data. The authors should report key metrics such as average error and failure rate across these new samples to provide a more comprehensive assessment of real-world applicability and generalization performance.

4.A broader comparison with alternative optimization methods—such as Bayesian optimization, Tree-structured Parzen Estimator (TPE), conditional GANs (cGANs), or diffusion models—is encouraged. Although implementing these approaches may be non-trivial, even a conceptual discussion or a limited experiment would help position the proposed method within the broader context of inverse design and optimization strategies. If experimental comparison is not feasible at this stage, the potential of these methods could be discussed as future work.

Author Response

Comment 1: 1.The rationale for adopting a custom shallow CNN architecture should be further clarified. Numerous well-established CNN models such as MobileNet, ShuffleNet, and ResNet18 have demonstrated strong performance in feature extraction and generalization. It is recommended that the authors implement and evaluate these baseline architectures within the proposed framework and perform a comparative analysis with their custom model. Such a systematic comparison would not only validate the proposed architecture but might also identify more effective alternatives, thereby enhancing model performance and the overall credibility of the work.

Response 1: New section 4.6 (Page 20) has been added where we adapted pretrained networks, specifically ResNet and MobileNet, for comparative analysis within our framework. This evaluation supports our decision to design a custom shallow CNN by demonstrating that it offers competitive performance due to the nature of our database and with significantly lower computational complexity, making it more suitable for our application constraints.

---------------------------------------------------------------------------------------------------

Commnet 2: The hyperparameter tuning strategy requires more detailed explanation. At present, the reported search space appears narrow and limited to a few basic parameters, without consideration of optimizer types or learning rate scheduling strategies. It is suggested that the authors include additional experiments that explore a broader set of hyperparameters. The application of advanced tuning methods such as Bayesian optimization—even in a limited experimental scope—would highlight the benefits of adaptive strategies and strengthen the technical rigor of the study.

Response 2: We used exhaustive grid search for hyperparameter tuning due to its simplicity, transparency, and suitability for our limited parameter space. This method ensured reproducibility and fair comparison across models while remaining computationally feasible given our resource constraints. Although more advanced techniques like Bayesian optimization offer efficient exploration, our goal was to establish a strong, interpretable baseline using key parameters such as learning rate, batch size, and network depth. Future work will explore broader hyperparameter spaces and adaptive tuning strategies to further optimize performance.

---------------------------------------------------------------------------------------------------

Commnet 3: The robustness of the proposed method should be validated using multiple magnetic field test samples. Currently, the evaluation includes only a single case, which may not sufficiently demonstrate the model’s generalizability and raises the concern of potential case selection bias. It is strongly recommended to test the model on multiple, preferably real-world, measurement datasets instead of relying solely on simulated data. The authors should report key metrics such as average error and failure rate across these new samples to provide a more comprehensive assessment of real-world applicability and generalization performance.

Response 3: The proposed method was used to design real planar coils that were printed and implemented in several sensors, as detailed in a separate publication. The applicability of the model is focused on planar coils, with the primary goal of translating spiral coil designs into equivalent planar versions by analyzing their magnetic field distribution and applying reverse engineering (Spiral Coil with core material to Planar coil with no core material). This translation process has significant economic value and has been successfully adopted by the company for practical sensor development.

---------------------------------------------------------------------------------------------------

Comment 4: A broader comparison with alternative optimization methods—such as Bayesian optimization, Tree-structured Parzen Estimator (TPE), conditional GANs (cGANs), or diffusion models—is encouraged. Although implementing these approaches may be non-trivial, even a conceptual discussion or a limited experiment would help position the proposed method within the broader context of inverse design and optimization strategies. If experimental comparison is not feasible at this stage, the potential of these methods could be discussed as future work.

Response 4: We appreciate suggestion and acknowledge the potential of advanced optimization methods such as Bayesian optimization, TPE, cGANs, and diffusion models for inverse design. While a direct comparison was beyond the scope of this study, we will consider incorporating and evaluating these approaches in our next publication, which will cover the fabrication of the coils and their validation through a dedicated experimental setup.

Reviewer 3 Report

Comments and Suggestions for Authors
  1. In abstract, please add more some important results.
  2. Do the authors write the principle or algorithm of the conventional neural networks. If no, please write it in the manuscript.
  3. Please review again about declare all parameters in all equations.
  4. Please review again about the reference(s) at important equations.
  5. In Fig. 6, why does the difference between calculation and simulation results?
  6. In Fig. 13, why does the difference between training and validation MAE?
  7. In Fig. 15, why does the difference between magnetic field input and output of the model?

Author Response

Comment 1:  In abstract, please add more some important results.  

Response 1: Updated accordingly.

---

Comment 2:  Do the authors write the principle or algorithm of the conventional neural networks. If no, please write it in the manuscript.  

Response 2: The neural network used in this work was designed from scratch. It is our original implementation tailored for the specific task.

---

Comment 3: Please review again about declare all parameters in all equations.

Response 3: Reviewed and updated accordingly. [line 216, 217]

---

Comment 4: Please review again about the reference(s) at important equations.  

Response 4: Reviewed. [Line 202]

---

Comment 5:  In Fig. 6, why does the difference between calculation and simulation results?

Response 5: The difference between the calculation and simulation results in Fig. 6 (Now it is Figure 7) is due to the mesh settings in the Simcenter Magnet FEM. After the first revision, the maximum mesh element size was reduced, resulting in more accurate FEM simulations with an error less than 2%. The updated results are presented in Fig. 7.

---

Comment 6:  In Fig. 13, why does the difference between training and validation MAE?  

Response 6: A small, consistent gap between training and validation MAE is normal because the model directly optimizes for the training data, making its error naturally slightly lower compared to the validation set. The validation MAE reflects generalization and typically follows a similar trend as the training MAE, just at a slightly higher level, indicating the model is learning effectively without overfitting.  Overfitting would only occur if the gap becomes larger. If the training MAE continues to decrease while the validation MAE starts rising, this means that the model is memorizing noise rather than generalizing. In our case, since both curves decrease and stabilize in parallel, the model shows good generalization.

---

Comment 7: In Fig. 15, why does the difference between magnetic field input and output of the model?

Response 7: The difference between the magnetic field input and the model output in Fig. 15 (Now Fig. 16) arises because the chosen values, drawn by hand, do not strictly follow the physical laws governing magnetic field distribution. Nonetheless, our work is powerful in that it can create a planar coil from a manually drawn magnetic field distribution, aiming to generate a magnetic field as close as possible to the desired specifications. 

Round 2

Reviewer 1 Report

Comments and Suggestions for Authors

The authors added needed clarifications, as suggested.

Author Response

All needed clarifications are added, as suggested by reviewer.

Reviewer 2 Report

Comments and Suggestions for Authors

1. How are the comparison parameters of the three models in Table 4 selected? Why are the comparison parameters of Learning rate, Epochs, and Batchsize inconsistent? It seems vague and needs further explanation from the author.
2. The conclusion is not written in a standardized way. It is recommended to list items and simplify the content.

Author Response

Comment 1: How are the comparison parameters of the three models in Table 4 selected? Why are the comparison parameters of Learning rate, Epochs, and Batchsize inconsistent? It seems vague and needs further explanation from the author.

Response 1: The main purpose of the comparison in Table 4 is to evaluate the best achievable performance of each model using the MAE and MSE metrics. The learning rate, number of epochs, and batch size shown in the table represent the optimal combination found for each model through grid search tuning (shown in Table 3). These hyperparameters were selected because they yielded the lowest error. Therefore, the comparison focuses on each model’s best performance (MAE, MSE), rather than on performance under identical training conditions (comparison of the best of each neural network).

---

Comment 2: The conclusion is not written in a standardized way. It is recommended to list items and simplify the content.

Response 2: The conclusion was revised to list key points and simplify the content for clarity and standardization [Page 23-24].